# Resistant *S. aureus* Isolates Capable of Producing Biofilm from the Milk of Dairy Cows with Subclinical Mastitis in Slovakia

Ján Király [1], Vanda Hajdučková [1], Gabriela Gregová [2,*], Tatiana Szabóová [2] and Emil Pilipčinec [1]

[1] Department of Microbiology and Immunology, The University of Veterinary Medicine and Pharmacy in Košice, 041 81 Košice, Slovakia; jan.kiraly@uvlf.sk (J.K.); vanda.hajduckova@uvlf.sk (V.H.); emil.pilipcinec@uvlf.sk (E.P.)

[2] Department of Public Veterinary Medicine and Animal Welfare, the University of Veterinary Medicine and Pharmacy in Košice, 041 81 Košice, Slovakia; tatiana.szaboova@uvlf.sk

\* Correspondence: gabriela.gregova@uvlf.sk

**Abstract:** *Staphylococcus* spp. is the most common cause of mastitis, with a significantly low cure rate. Bacterial characteristics like adhesion and biofilm formation, as well as extracellular factors, can affect the pathogenesis of staphylococcal mastitis. The study's objectives were to confirm *S. aureus*, assess their antibiotic resistance, identify methicillin resistance genes, verify biofilm formation, and detect biofilm-associated genes from bovine mastitis samples using multiplex PCR (mPCR). From 215 milk samples, six were confirmed as *S. aureus*. Most isolates were sensitive to all measured antibiotics. One isolate was identified as an inducible form of MLSB resistance (macrolides, lincosamides, and streptogramin B resistance), while the other two isolates were resistant to penicillins and carboxypenicillins. In *S. aureus* cultures used for methicillin resistance genotypic analysis by PCR, the *mecA* and *mecC* genes were not found. Biofilm formation phenotypes were determined in four strains. An mPCR analysis revealed that all strains of *S. aureus* carried *icaABCD*, *agrA*, *srtA*, *fnbA*, *clfA*, and *clfB* genes. Only in one isolate was the *fnbB* gene detected; the bap gene was not detected in any of the isolates. This emphasizes the importance of using appropriate treatment and continuous monitoring of *S. aureus* to prevent the spread of antibiotic-resistant strains in dairy cow farms.

**Keywords:** subclinical mastitis; *Staphylococcus aureus*; biofilm; multiplex PCR; MIC



## 1. Introduction

Mastitis caused by bacterial infections is the most common disease in dairy cows. It is the reason for antimicrobial treatment in dairy cows and a public health threat. It occurs in clinical or subclinical form. The pathogenic potential of bacteria, an individual's immunity, and their overall health all influence how severe mastitis can become. Subclinical mastitis is an asymptomatic infection that influences milk production and milk quality [1]. The main problem connected with the subclinical form of mastitis is its higher incidence, longer persistence, and speed of spread in dairy farming. A commonly used diagnostic method for detecting the subclinical form of mastitis is a test to determine the somatic cell count in milk [2,3].

Dairy cow mastitis is frequently caused by the bacterial genera *Staphylococcus*, *Streptococcus*, and *Escherichia*. The predominant infectious pathogens causing subclinical mastitis with a significantly low cure rate are staphylococcal infections [4]. Additionally, *S. chromogenes*, *S. warneri*, and *S. xylosus* were isolated from animals that exhibit clinical mastitis, which is defined by mild, moderate, or severe symptoms as well as chronic infections [5]. Treatment of clinical mastitis should follow national and international prudent use guidelines. The systemic (parenteral) route of mastitis treatment is more efficient than the intramammary route because antimicrobials can better penetrate udder tissue. For the treatment of clinical mastitis, broad-spectrum antibiotics (oxytetracycline, macrolides, trimethoprim-sulfonamide, and ceftiofur combination) are frequently used [6,7].

The consumption of veterinary antibiotics decreased in Slovakia during the last decade. The top-selling antimicrobials include tetracyclines (46.0%), penicillins (14.0%), and sulfonamides (52.0%) [8]. Of the total sales of antimicrobials, approximately 41.0% were used in beef farms, 42.0% in swine farms, 10.0% in turkey farms, 3.0% in chicken farms, and about 4.0% in other farms. Tetracyclines account for the highest sales volume of ATB used in food animals (4,117,031 kg in 2019), which is 67.0%. They are followed by penicillins at 12.0%, macrolides at 8.0%, sulfonamides at 5.0%, aminoglycosides at 5.0%, lincosamides at 2.0%, cephalosporins at less than 1.0%, and fluoroquinolones at less than 1.0%. About 81.0% of cephalosporins, 65.0% of sulfonamides, 45.0% of aminoglycosides, and 42.0% of tetracyclines were used in cattle production. In pig production, approximately 85.0% of lincosamides and 40.0% of macrolides were utilized. For use on turkey farms, 66.0% of penicillins were used [9].

*S. aureus* is the most important causative agent of subclinical mastitis worldwide, which is also highly isolated from milk samples from dairy cows with mastitis. In addition, it is the most common isolated foodborne pathogen, while milk and milk products are potential sources of infection [10]. Both extracellular elements and bacterial characteristics, like adhesion and biofilm formation, can have an impact on the pathogenesis of staphylococcal mastitis. The expression of virulence factors is upregulated during bacterial infection, leading to increased resistance to phagocytosis and upregulation of genes through which host tissue destruction occurs [1,11,12] *S. aureus* is able to escape the innate and adaptive immunity of the host and resist the effects of β-lactam antibiotics. The long-term use of various antibiotics leads to an increased prevalence of multiresistant strains [13]. Methicillin-resistant *S. aureus* (MRSA) was initially detected as a hospital-acquired infection (HA-MRSA), later in the human community (CA-MRSA), and more recently in livestock (LA-MRSA) [14]. Bacteria carrying the *mecA* gene and its homolog *mecC* can encode a modified low-affinity penicillin-binding protein, rendering them resistant to methicillin and most other beta-lactams. Both genes are transferred on mobile genetic elements (MGEs) called staphylococcal chromosome cassette—mec (SCC *mec*) [15], which contains regulation genes *mecI* (encoding repressor) and *mecR1* (encoding sensory protein). Several structural variants of SCC *mec* were described that differ in their genetic content, size, and structural organization. Twelve major types of SCC *mec* elements in MRSA strains were classified, some of which were divided into subtypes [16]. SCC *mec* element typing is essential because, in combination with *S. aureus* chromosome genotype, SCC *mec* type is an important characteristic for defining MRSA clones in epidemiological studies and for understanding the evolution of these clones. Strains containing different elements of SCC *mec* differ in their sensitivity to antibiotics, which is significant from the point of view of clinical consequences. A divergent homolog of the *mecA* gene, described as *mecC*, was discovered in the same SCC *mec* element (SCC *mec* type XI) in the genome of *S. aureus* strain LGA251, which was isolated from bovine mastitis [17].

In recent years, there has been an increase in the occurrence of MRSA in different samples. It resulted in increased use of alternative antibiotics (macrolides, lincosamides, and streptogramin B) for the treatment of staphylococcal infections. Because of its good pharmacokinetic properties, proven effectiveness, low cost, oral and parenteral form availability, good tissue penetration, good abscess accumulation, inhibition of staphylococcal toxin production, and lack of need for renal dosing modifications, clindamycin is the most commonly used antibiotic in this class [6].

Biofilm-forming *S. aureus* strains are very resistant to antibiotics [18,19], which complicates the eradication of biofilm infections [20,21]. Biofilm consists of several bacterial layers protected by the exopolysaccharide glycocalyx. As one of the virulence factors, it facilitates the adherence and colonization of bacteria in the epithelium of the mammary glands. It contributes to protection against the immune response, decreases eradication of pathogens, and results in recurrent or persistent infections [22]. The mechanism of formation of biofilm by staphylococci is a complicated process involving many genes and regulation factors. Its formation basically consists of the initial adhesion of bacteria to the surface, the formation

of microcolonies, the maturation of the biofilm, and dispersion. In biofilm-forming *S. aureus*, several types of adhesive molecules used by bacteria to attach to the host tissue were identified. Adhesion to biotic surfaces is mediated by a group of surface-exposed proteins expressed in the *S. aureus* cell wall, which are referred to as MSCRAMMs (microbial surface components recognizing adhesive matrix molecules) [23]. The attachment to extracellular proteins of cells in *S. aureus* is mediated by fibronectin-binding proteins (*FnbA*, *FnbB*) and fibrinogen-binding proteins (*ClfA*, *ClfB*). In strains isolated from bovine mastitis, in strong producers of biofilm, an important role of biofilm-associated protein (Bap) (also belonging to the MSCRAMM group) was identified [24] during the primary attachment and accumulation of bacterial cells. It was also observed that formation of biofilm mediated by Bap is independent of production of polysaccharide intracellular adhesion (PIA). A significant characteristic of the majority of MSCRAMMs is the amino acid Leu-Pro-X-Thr-Gly (LPXTG) motif, which is recognized and cleaved by enzyme sortase A, which catalyzes the covalent bonding of MSCRAMMs to the peptidoglycan layer of bacteria during assembly of the cellular wall at vegetative growth. Excessive expression of gene *srtA* (encoding sortase A) increases the rapidity of anchorage of surface proteins during cellular wall biosynthesis and contributes to increased virulence [25]. During the phase of multiplication of bacteria in the biofilm resulting in development of a number of bacterial layers, a staphylococcal exopolysaccharide referred to as PIA (polysaccharide intercellular adhesion) is produced on the basis of function, or poly-$\beta$-1-6-N-acetylglukózamín (PNAG) on the basis of its chemical character. The genes necessary for biosynthesis of PIA are encoded in the locus of intracellular adhesion (*ica*), and this locus is the primary determinant supporting adhesive interactions between bacterial cells [26]. Products of the genes of locus *icaABCD* are inevitable for formation of biofilm, and products of genes *icaA* and *icaD* play a key role in this process [27].

## 2. Materials and Methods

### 2.1. Sampling, Isolation, and Identification of Staphylococcus aureus

Milk samples (*n* = 215) were collected from dairy herds in Slovakia with reduced milk quality and milk production. Milk samples from dairy cows with subclinical mastitis were positive for the NK-test (natural killer test) (Bioveta, Ivanovice na Hané, Czech Republic). The dairy cows were without clinical signs of mastitis, without previous cases of mastitis in the same lactation, and without treatment. The NK test is a rapid determination used to find dairy cows suspected of having mastitis based on an increased somatic cell count and acid pH in milk. The somatic cell counts of 100,000–300,000 cells/mL (mild coagulation), 300,000–500,000 cells/mL (coagulation with mild gel formation), and 500,000–1,500,000 cells/mL (strong precipitation with gel formation) indicate a positive reaction.

Isolates were stored at −80 °C in Microbank cryotubes (Pro-Lab, Mississauga, ON, Canada). Isolation and identification of *S. aureus* was performed by classical microbiological methods. Hemolytic activity was determined using blood agar, and Baird–Parker agar was used to evaluate lecithinase activity. Phenotypic identification of the isolates was performed using the STAPHYTest 24 biochemical series (Erba Lachema, Brno-Řečkovice a Mokrá Hora, Czech Republic). Subsequently, the isolates were subjected to confirmation at the molecular level by using mPCR.

### 2.2. DNA Extraction

From the overnight culture of isolates, genomic DNA was extracted by modified Brain Heart Infusion Broth (mBHI, HiMedia Laboratories, Mumbai, India) containing 1.0% glucose and 2.0% NaCl using the High Pure PCR Template Preparation Kit (Roche Molecular Systems, Inc., Pleasanton, CA, USA). The amount and purity of DNA were determined using an ND-8000 spectrophotometer (ThermoFisher SCIENTIFIC, Waltham, MA, USA).

### 2.3. Gene Detection Using Simplex PCR (PCR) and Multiplex PCR (mPCR)

For the specification of *S. aureus* by mPCR, we used primers amplifying the *Staphylococcus* genus 16S rRNA gene segment and species-specific primers for *S. aureus* that detect the *eap* (extracellular adhesion protein) and *nuc* (thermostable nuclease) genes (Table 1).

**Table 1.** Primers used for PCR analysis.

| Gene | Primer | Sequence (5′→3′) | Product (bp) | Reference |
|---|---|---|---|---|
| *16S rRNA* | 16S rRNA_Fw<br>16S rRNA_Rev | CTACAATGGACAATACAAAGGGC<br>TCACCGTAGCATGCTGATCT | 141 | [28] |
| *eap* | EAP-CON1<br>EAP-CON2 | TACTAACGAAGCATCTGCC<br>TTAAATCGATATCACTAATACCTC | 230 | [29] |
| *nuc* | nuc_Fw<br>nuc_Rev | ACCTGCGACATTAATTAAAGCG<br>TGTTTCAGGTGTATCAACCAATAATAG | 103 | This study |
| *mecA* | mecA_Fw<br>mecA_Rev | TGGAAGTTAGATTGGGATCATAGC<br>CGATGCCTATCTCATATGCTGTT | 154 | This study |
| *mecC* | mecC_Fw<br>mecC_Rev | GACGATGGATCTGGTACAGCA<br>CATTCATGAATGGATAAACATCGTA | 94 | This study |
| *bap* | bap_Fw<br>bap_Rev | TTGACGAGGTTGGTAATGGC<br>CGCCTACAGTTTCTGGTAATGC | 87 | This study |
| *icaA* | icaA_Fw<br>icaA_Rev | CTTGCTGGCGCAGTCAATAC<br>GTAGCCAACGTCGACAACTG | 75 | [30] |
| *icaB* | icaB_Fw<br>icaB_Rev | ATACCGGCGACTGGGTTTAT<br>ATGCAAATCGTGGGTATGTGT | 141 | [31] |
| *icaC* | icaC_Fw<br>icaC_Rev | CTTGGGTATTTGCACGCATT<br>GCAATATCATGCCGACACCT | 209 | [32] |
| *icaD* | icaD_Fw<br>icaD_Rev | ACCCAACGCTAAAATCATCG<br>GCGAAAATGCCCATAGTTTC | 211 | [31] |
| *srtA* | srtA_Fw<br>srtA_Rev | GTGGTACTTATCCTAGTGGCAGC<br>GCCTGCCACTTTCGATTTATC | 183 | [33] |
| *agr* | agr_Fw<br>agr_Rev | TCGTAAGCATGACCCAGTTG<br>AAATCCATCGCTGCAACTTT | 96 | [33] |
| *fnbA* | fnbA_Fw<br>fnbA_Rev | GAAGTGGCACAGCCAAGAAC<br>ACGTTGACCAGCATGTGG | 192 | [33] |
| *fnbB* | fnbB_Fw<br>fnbB_Rev | CAATGATCCTATCATTGAGAAGAGTG<br>CCTTCTACACCTTCAACAGCTGTA | 156 | [33] |
| *clfA* | clfA_Fw<br>clfA_Rev | GAGAGCATTTAGTTTAGCGGCA<br>TCACCTTTAACAGCAGAATTAGGC | 180 | This study |
| *clfB* | clfB_Fw<br>clfB_Rev | GTCTACACAAACGAGCAATACCAC<br>TGAGGAACAGTTTGATCTTGCA | 120 | This study |

The presence of genes responsible for antibiotic resistance was monitored by PCR using primers amplifying sections of the *mecA* and *mecC* genes. Genes associated with biofilm were monitored by PCR (*bap*) and by mPCR (*icaABCD*, *srtA*, and *agrA*). A combination of primers detecting genes *icaD*, *srtA*, and *agrA*, as well as primers detecting *icaA*, *icaB*, and *icaC*, were chosen for mPCR based on the size of the products. Two primer combinations were used in mPCR: *fnbA* with the *fnbB* gene and *clfA* with the *clfB* gene. The primer sequences are shown in Table 1.

The PCR and mPCR reaction mixture was contained in the resulting volume 0.5 ng of isolated DNA, 1x DreamTaq Green PCR Master Mix (ThermoFisher SCIENTIFIC, USA), 0.125 pmol (*16S rRNA*), 0.5 pmol (*eap*), 0.25 pmol (*nuc*), 0.5 pmol (*mecA*, *mecC*), 0.5 pmol (*bap*), 0.25 pmol (*icaB*, *icaC*, *icaD*, *agrA*, *srtA*), 0.375 pmol (*icaA*), 0.5 pmol (*fnbB*), and 0.25 *fnbA*, *clfA*, *clfB*) of each primer ("forward, reverse") and water. The PCR and mPCR reaction took place on a Mastercycler® nexus X2 thermocycler (Eppendorf, Hamburg, Germany) under the optimized conditions listed in the tables (Tables 2 and 3).

**Table 2.** Conditions of mPCR detection of 16S rRNA and genes *eap* and *nuc*.

| Process | Temperature (°C) | Time (s/min) | Number of Cycles |
|---|---|---|---|
| Initial denaturation | 95 | 3 min | 1 |
| Denaturation | 95 | 30 s | |
| Annealing | 55 | 30 s | 30 |
| Extension | 72 | 20 s | |
| Denaturation | 95 | 30 s | |
| Annealing | 61 | 30 s | 10 |
| Extension | 72 | 20 s | |
| Final extension | 72 | 10 min | 1 |

**Table 3.** Conditions for PCR detection of genes *mecA*, *mecC*, and *bap* and for mPCR detection of genes *icaABCD*, *srtA*, *agrA*, *fnbA/B*, and *clfA/B*.

| Process | Temperature (°C) | Time (s/min) | Number of Cycles |
|---|---|---|---|
| Initial denaturation | 95 | 3 min | 1 |
| Denaturation | 95 | 30 s | |
| Annealing | 61 | 30 s | 35 |
| Extension | 72 | 20 s | |
| Final extension | 72 | 10 min | 1 |

Amplified DNA sections were separated electrophoretically (Wide Mini-Sub® GT Cell, BIORAD, Hercules, CA, USA) on a 2.5% agarose gel using a non-toxic nucleic acid detection reagent called GoodView™ (Amplia, SR, Bratislava, Slovakia). PCR products were visualized in the gel after electrophoresis using the UV light of an Ultraviolet Transilluminator (Bio-Imaging Systems, Modi'in-Maccabim-Re'ut, Israel) and recorded using a Kodak Gel Logic 100 Digital Imaging System (Kodak, Rochester, NY, USA).

### 2.4. Antibiotic Susceptibility Testing

*S. aureus* isolates confirmed by PCR were analyzed for antibiotic susceptibility. Minimum inhibitory concentration (MIC) for determination of phenotype antibiotic resistance was determined by a microdilution colorimetric method according to CLSI VET01-S2 [34] and EUCAST [35] using the Miditech system (Bratislava, Slovakia) with an interpretive reading of MIC. The following antibiotics were tested: ampicillin (AMP), ampicillin + sulbactam (SAM), oxacillin (OXA), cefoxitin (FOX), piperacillin + tazobactam (TZP), erythromycin (ERY), clindamycin (CLI), linezolid (LNZ), rifampicin (RIF), gentamicin (GEN), teicoplanin (TEC), vancomycin (VAN), trimethoprim (TMP), chloramphenicol (CHL), tigecycline (TGC), moxifloxacin (MFX), ciprofloxacin (CIP), tetracycline (TET), trimethoprim + sulphonamide (COT), and nitrofurantoin (NIT). The antibioticogram profile and MIC profile will serve to determine mechanisms of resistance (MRCoNS, MRSA, MLSB) [36].

### 2.5. Biofilm Activity Testing—Crystal Violet Biofilm Testing

Biofilm activity was tested on a 96-well polystyrene microtiter plate by staining with crystal violet, using a modified colorimetric method according to O'Toole et al. [37]. The tested strains of *S. aureus* were incubated as pure bacterial cultures at 37 °C overnight (for 18 h) on blood agar. From individual bacterial colonies and physiological solution, one McFarland turbidity was suspended; 100 µL of the suspension was added to the wells of a polystyrene plate with 100 µL of modified Brain Heart Infusion Broth (mBHI, HiMedia Laboratories, Mumbai, India) with 1.0% glucose and 2.0% NaCl. The reference strains were the non-biofilm-forming *S. epidermidis* CCM 4418 and the biofilm-forming *S. aureus* CCM 4223 (Czech Collection of Microorganisms, Brno, Czech Republic). The negative control was pure broth.

After being incubated for 24 h at 37 °C, the planktonic cell medium was removed, and the wells were washed four times with distilled water. Subsequently, the biofilms were

stained by adding 200 μL of 0.1% crystal violet solution (Merck, Darmstadt, Germany) and incubated for 30 min at room temperature. Following four rounds of washing and ten minutes of room-temperature drying, the adhered cells were dye-free by adding a 30.0% glacial acetic acid solution in a volume of 200 μL per well. The optical density was determined spectrophotometrically by measuring the absorbance at a wavelength length of 550 nm using a SYNERGY READER 4 plate reader (BioTek, Merck, Germany).

### 2.6. Statistical Evaluation of Biofilm Formation

Biofilm formation was evaluated based on the obtained data by statistical analysis using the GraphPad Prism 6.01 program (GraphPad Inc., San Diego, CA, USA) using one-way analysis of variance (ANOVA) and Tukey's test to determine significance at $p < 0.001$.

## 3. Results

### 3.1. Confirmation of the Genus S. aureus

*S. aureus* isolates ($n = 11$) from milk samples ($n = 215$) from dairy cows with a subclinical form of mastitis were identified using traditional microbiological methods. *Staphylococcus* spp. was confirmed via mPCR using the 16S rRNA sequence (141 bp), *nuc* (103 bp), and *eap* genes (230 bp).

All field isolates identified using standard microbiological procedures had positive detections of the 16S rRNA gene specific to the *Staphylococcus* genus. Six isolates (isolates no. 7 to 12) had species-specific genes for *S. aureus nuc* and *eap* (Figure A1).

### 3.2. Antimicrobial Susceptibility Profile

The MIC (minimal inhibition concentration) of antibiotics was determined in six confirmed *S. aureus* isolates (Table 4).

*S. aureus* isolates were mainly sensitive to all monitored antibiotics. One isolate was resistant to ampicillin and ampicillin with sulbactam. One strain showed resistance to erythromycin, clindamycin, and ampicillin. One of the two isolates, which exhibited resistance to both penicillins and carboxypenicillins, was phenotypically identified as an inducible type of MLSB-resistant *S. aureus* (macrolide-lincosamide-streptogramin B) (Figure A2).

None of the confirmed *S. aureus* isolates from subclinical mastitis cows showed resistance to cefoxitin or oxacillin. The significance of this for animal health lies in the fact that β-lactam antibiotics remain one of the most widely used antimicrobial therapeutics for treating mastitis, and there are few other options for treating the condition with non-β-lactam antibiotics [10].

To confirm the obtained MIC of *S. aureus* sensitivity to antibiotics by determining the MIC, PCR detection of the genes *mecA* and *mecC* was performed. The presence of specified genes was not detected, which correlates with our MIC results.

It demonstrates that phenotypic testing of resistance can result in both false positives and false negatives for MRSA. Most of the published studies aimed at monitoring the susceptibility of *S. aureus* strains to antibiotics report results obtained only through phenotypic resistance testing without demonstrating the presence of the *mecA* and *mecC* genes responsible for the production of an altered penicillin-binding protein 2a (PBP2a), which has a lower affinity for β-lactam antimicrobial substances than normal PBP [10].

**Table 4.** MIC results of *S. aureus* isolates and resistance assessment.

| Sample | 7 | | 8 | | 9 | | 10 | | 11 | | 12 | |
|---|---|---|---|---|---|---|---|---|---|---|---|---|
| ATB | MICxG (mg/L) | Antimicrobial Susceptibility Profile | MICxG (mg/L) | Antimicrobial Susceptibility Profile | MICxG (mg/L) | Antimicrobial Susceptibility Profile | MICxG (mg/L) | Antimicrobial Susceptibility Profile | MICxG (mg/L) | Antimicrobial Susceptibility Profile | MICxG (mg/L) | Antimicrobial Susceptibility Profile |
| AMP | 0.25 | S | 0.25 | S | 0.25 | S | 2 | R | 0.25 | S | >32 | R |
| SAM | 0.25 | S | 0.25 | S | 0.25 | S | 0.5 | S | 0.25 | S | >32 | R |
| OXA | 0.25 | S | 0.25 | S | 0.25 | S | 0.5 | S | 0.25 | S | 0.25 | S |
| FOX | 2 | S | 2 | S | 2 | S | 2 | S | 2 | S | 2 | S |
| TZP | 0.5 | S | 0.5 | S | 0.5 | S | 2 | S | 0.5 | S | 0.5 | S |
| ERY | 0.25 | S | 0.25 | S | 0.25 | S | >8 | R | 0.25 | S | 0.12 | S |
| CLI | 0.12 | S | 0.12 | S | 0.12 | S | 4 | R | 0.12 | S | 0.06 | S |
| LNZ | 4 | S | 4 | S | 4 | S | 2 | S | 4 | S | 2 | S |
| RIF | 0.03 | S | 0.03 | S | 0.03 | S | 0.03 | S | 0.03 | S | 0.03 | S |
| GEN | 0.5 | S | 0.5 | S | 1 | S | 1 | S | 0.5 | S | 0.5 | S |
| TEC | 1 | S | 2 | S | 2 | S | 2 | S | 1 | S | 1 | S |
| VAN | 1 | S | 2 | S | 1 | S | 1 | S | 1 | S | 1 | S |
| TMP | 2 | S | 4 | S | 4 | S | 2 | S | 1 | S | 1 | S |
| CHL | 8 | S | 8 | S | 8 | S | 16 | R | 8 | S | 8 | S |
| TGC | 0.12 | S | 0.06 | S | 0.06 | S | 0.12 | S | 0.06 | S | 0.06 | S |
| MFX | 0.12 | S | 0.06 | S | 0.06 | S | 0.06 | S | 0.03 | S | 0.03 | S |
| CIP | 0.5 | I | 0.25 | I | 0.5 | I | 0.5 | I | 0.25 | I | 0.25 | I |
| TET | 1 | S | 0.5 | S | 0.25 | S | 0.5 | S | 0.25 | S | 0.25 | S |
| COT | 0.25 | S | 0.25 | S | 0.25 | S | 0.25 | S | 0.25 | S | 0.25 | S |
| NIT | 16 | S | 32 | S | 32 | S | 16 | S | 16 | S | 16 | S |

S—sensitive; I—intermediate; R—resistant; MICxG—geometric mean MIC.

### 3.3. Phenotypic Identification of Biofilm-Forming S. aureus

The biofilm formation ability of individual *S. aureus* isolates was tested using a modified colorimetric method according to O'Toole et al. [37]. Statistically, 4 out of 6 isolates of *S. aureus* (samples 8, 10, 11, and 12) were evaluated as producers of biofilm, as well as a positive control (reference strain *S. aureus* CCM 4223). Two isolates (samples 7 and 9) showed a significantly lower ability to produce biofilm (Figure 1).

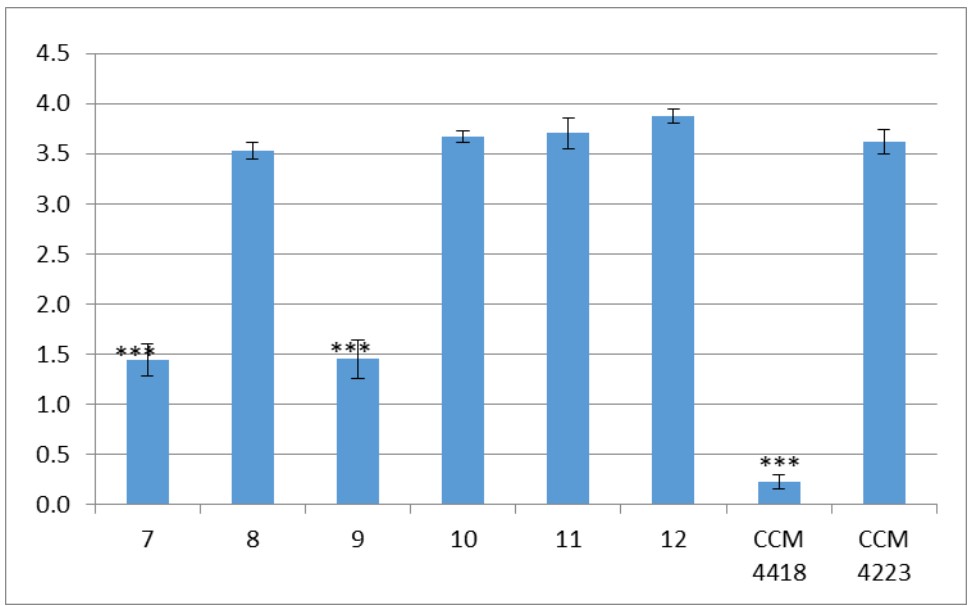

**Figure 1.** Biofilm formation in *S. aureus* isolates from subclinical mastitis milk. CCM 4418—non-forming biofilm *S. epidermidis* (negative control), CCM 4223—biofilm forming *S. aureus* (positive control), Samples 7–12—strains with weak or strong production of biofilm. *** significantly lower ability to produce biofilm ($p < 0.005$).

Growth conditions influence biofilm formation both in vitro and in vivo, so phenotypic and genotypic analysis of biofilm-associated genes is required to identify strains that form biofilms. For this reason, in every clinical isolate of *S. aureus*, we also monitored the biofilm-associated genes by using the simplex and multiplex PCR methods (Figures A3–A6).

### 3.4. Detection of Biofilm-Associated Genes in S. aureus

By mPCR analyzing the amplified segments of the *icaABCD*, *srtA*, and *agrA* genes, their presence was confirmed in all (100%) *S. aureus* isolates (isolates no. 7 to 12) in our study (Figures A3 and A4).

Genes that encode proteins of the MSCRAMM group of adhesins (LPXTG proteins), which *S. aureus* uses to bind to extracellular proteins in mammals, are another set of genes involved in the biofilm formation process. Among the most significant biofilm-associated genes, *fnbA* and *fnbB* encode the fibronectin-binding proteins FnbA and FnbB, and the clfA and clfB genes encode the fibrinogen-binding proteins referred to as clumping factors A (ClfA) and B (ClfB). The MSCRAMM group of adhesins also includes the biofilm-associated protein (Bap), which supports the initial and intercellular adhesion of *S. aureus* strains. The bap gene was detected mainly in strong biofilm producers of *S. aureus* isolated from cattle [38]. Using the mPCR method, we monitored the presence of a pair of *fnbA*, *fnbB*, and *clfA*, *clfB* genes in clinical isolates (Figures A5 and A6) and the bap gene by simplex PCR. In every isolate, the *fnbA*, *clfA*, and *clfB* genes were found (isolates no. 7 to 12). The presence of the *fnbB* gene was recorded only in one isolate (no. 10), and there was no detection of the bap gene in any clinical strain of *S. aureus*.

## 4. Discussion

Based on an investigation of 813 isolates, which included species of the genus *Staphylococcus* in addition to *S. aureus* [29], the *eap* gene was detected in 597 isolates of *S. aureus*. In 216 staphylococcal isolates, which included 47 various species (coagulase-negative, other coagulase-positive, or coagulase-variable subspecies of staphylococci), the *eap* gene was not detected. Based on the results obtained from transcriptional and protein analyses, they proved that species other than *S. aureus* do not express *eap* homologues. Therefore, they established the sensitivity and specificity of the newly developed PCR aimed at detecting the *eap* gene at 100%. For the molecular detection of *S. aureus* through the amplification of part of the *nuc* gene, Gonzalez-Dominguez et al. [39] also used it in their work. As in our study, they used specific primers designed to identify a segment within sequence homologies shared between the *nuc* gene of *S. aureus* species. The authors of the study state that despite the high specificity of the Staph-API kit (92.49%) to diagnose *Staphylococcus* species, the identification by means of phenotyping of staphylococci obtained from clinical samples does not reach the stated specificity. This may be one of the explanations for our results. We confirmed *S. aureus* in 6 out of 11 isolates by the PCR method, which represents 54.5%. Qolbaini et al. [40] obtained similar results. They also detected the *nuc* gene in 49 (57.0%) of 86 *S. aureus* isolates identified by classical microbiological methods. For proper identification, it is necessary to perform sequence analysis of isolates without *eap* and *nuc* genes.

*S. aureus* isolates from the milk of dairy cows with a subclinical form of mastitis were mostly sensitive to all analyzed antibiotics. Based on its phenotypic characteristics, one strain of *S. aureus* was identified as MLSB-resistant *S. aureus*, which can be inducible. MRSA was not identified by using genotypic or phenotypic methods. The MLSB (macrolide-lincosamide-streptogramin B) inducible resistance type of *S. aureus* can encode over 60 genes, such as rRNA methylases (*ermA-ermZ*, *erm(30)-erm(42)*), efflux pumps (*isaA-isaC* genes, *mefA-mefG*, *msrA*, *vgaA-vgaC*) and inactivating genes encoding esterases, lyases, phosphorylases, and transferases (*ereA*, *ereB*, *ereC*, *lnuA-lnuF*, *vatA-vatG*) [41].

The widespread use of the MLSB antibiotics (mostly clindamycin) has increased the number of MLSB-resistant *S. aureus* isolates [42]. The rate of MLSB-resistant staphylococci varies between countries and species. Many authors report the high occurrence of MLSB in the human community, which causes nosocomial diseases. On the other hand, a high rate of erythromycin-resistant staphylococci was also observed in veterinary practice [43].

Szczuka et al. [44], from Poland, investigated the presence of genes linked to antibiotic resistance in milk samples and milk products. Antibiotic-resistant genes, which are critical for both human and veterinary medicine, were presented in *S. aureus* strains, i.e., β-lactams (*mecA*) and aminoglycosides. According to the antimicrobial susceptibility test, 74.0% of the strains of *S. aureus* were resistant to at least one of the 16 tested antibiotics, representing 11 different categories. Furthermore, 28.0% of the strains were multidrug-resistant, and two MRSAs demonstrated notable antibiotic resistance. Their research indicates that *S. aureus* strains expressing enterotoxin and antibiotic resistance genes are present in dairy products. In foods originating from animals, they detected MRSA strains as well as MSSA isolates that demonstrated multidrug resistance [44].

Cvetnić et al. [45] reported that ten MRSA strains (4.2%) were isolated from milk samples of cows with subclinical mastitis in Croatia. In a Norwegian study [46], they found that all *S. aureus* strains isolated from milk and cheese were sensitive to the 12 antibiotics tested. The Chinese research found that raw milk samples had a high MRSA frequency of 51.6%, and 80.6% of *S. aureus* strains identified from milk were resistant to at least one antibiotic [47]. Moreover, in our former study [48], the highest antimicrobial resistance to penicillin (91.0%) and erythromycin (67.0%) was confirmed in *S. aureus* and CoNS isolates from sheep and goat cheeses.

According to our results, the prevalence of *S. aureus* strains isolated from the milk of dairy cows with subclinical mastitis capable of forming a biofilm (66.7% of isolated *S. aureus* had the ability to form biofilm) is similar to that in the study of Rychshanova et al. [49]. The

achieved results highlight the need for increased attention in relation to the prevalence of biofilm-forming strains isolated from cattle farms due to the existence of a potential risk of the spread of these strains and the contamination of milk and milk products with an impact on public health. Wang et al. [50] emphasize the importance of biofilm in antibiotic resistance gene transfer. Ineffective treatment of mastitis caused by biofilm-forming *S. aureus* with antibiotics can increase the risk of antibiotic resistance, which poses a threat to human and animal health [1].

PIA encoding the intercellular adhesion locus (*ica*) formed from the *icaABCD* genes, of which the *icaA* and *icaD* genes play the most important role in the process of biofilm formation, is reported as an important virulence factor in the pathogenesis of mastitis [51]. Sortase A is a transpeptidase enzyme that facilitates the attachment of numerous surface proteins involved in bacterial cell wall synthesis. These proteins are associated with aspects of virulence, such as invasibility and adhesion. *S. aureus* strains lacking the *srtA* gene are defective in the anchoring of LPXTG proteins in the cell wall; therefore, bacteria are unable to attach cell surface proteins, which are necessary for adhesion to eukaryotic cell structures [52]. The agr locus plays an important role in controlling the expression of most *S. aureus* virulence factors. The *agrA* gene encodes a response regulator that is part of the *agr* quorum-sensing system and, by binding to DNA, activates the P3 promoter, thereby regulating the expression of various virulence factors, the dispersion phase, and the survival of *S. aureus* in the host tissue [26].

The prevalence of the genes detected by us (*icaABCD*, *srtA*, *agrA*) is different in published studies [22,26,38,51]. Studies focused on the detection of genes encoding the proteins *FnbA*, *FnbB*, *ClfA*, *ClfB*, and *Bap* also do not agree in their results of determining the percentage of prevalence of individual genes [22,26,32,53]. Point mutations in the corresponding genes may be the reason for variations in gene detection in biofilm-forming strains. Therefore, differences in DNA sequences are the cause of unsuccessful amplification of gene segments in some clinical isolates, and the results obtained are likely to show false negativity. The two most studied mechanisms responsible for biofilm formation are the PIA-dependent biofilm formation mechanism under the control of the *ica* operon and the PIA-independent biofilm formation mechanism mediated by the Bap protein [38]. Results in published studies showing the absence of genes involved in the process of biofilm formation in biofilm-positive *S. aureus* strains may be related to biofilm formation under the control of other genes responsible for biofilm formation [26]. However, even in biofilm-negative isolates, we found that 100% of the monitored genes (*icaABCD*, *srtA*, *agrA*, *fnbA*, *clfA*, and *clfB*) were present in our work, with the exception of *fnbB* (16.7%) and *bap* (0%). This may be related to the different abilities of *S. aureus* isolates to form biofilm in vivo and in vitro.

## 5. Conclusions

Staphylococci, and especially *S. aureus*, are important pathogens in humans and animals. The overall occurrence of *S. aureus* is variable and varies between farms and regions. Based on our results, *S. aureus* is not the most common pathogen of the mammary gland. Inaccuracies in the identification of *S. aureus* in clinical samples based on classic microbiological procedures point to the need to also include methods for the detection of species-specific *nuc* and *eap* genes in the diagnostic procedure. The overall rate of MRSA in dairy farms is low, but there is still a risk of it increasing. By monitoring sensitivity to antibiotics, no MRSA strain was recorded, which is a positive sign since the antibiotics of first choice in the treatment of staphylococcal mastitis are β-lactams. Continuous monitoring of antibiotic resistance is essential because resistant strains of *S. aureus* (MLSB and penicillin-resistant strains) are emerging, which poses a problem in the treatment of bovine mastitis. The obtained results of the detection of MRSA strains were also confirmed by the detection of *mecA* and *mecC* genes, which were not present in any *S. aureus* isolate.

The results of our study show a relationship between the occurrence of *S. aureus* isolates in the milk of dairy cows with subclinical mastitis and biofilm formation, which points to the pathogenic potential of these strains.

The occurrence of strains capable of forming a biofilm in a herd of dairy cows points to the need to pay attention to the observance of good milking practices and to implement procedures for effective prevention, control, and treatment of mammary gland infection.

**Author Contributions:** Conceptualization, J.K. and G.G.; methodology, J.K., G.G., V.H. and T.S.; validation, J.K. and G.G.; formal analysis, J.K., G.G. and E.P.; investigation, J.K., G.G., V.H. and T.S.; resources, J.K. and G.G.; data curation, J.K. and G.G.; writing—original draft preparation, J.K. and G.G.; writing—review and editing, J.K. and G.G.; visualization, J.K. and G.G.; supervision, E.P.; funding acquisition, G.G. and E.P. All authors have read and agreed to the published version of the manuscript.

**Funding:** This publication was supported by the Slovak Research and Development Agency under the contract no. APVV-15-0377 and by the cultural and educational grant agency KEGA of the Ministry of Education, Science, Research and Sports of the Slovak Republic, project numbers KEGA 007UVLF-4/2021 and KEGA 001UVLF-4/2022.

**Institutional Review Board Statement:** All procedures involving animals followed the guidelines stated in the Guide for the Care and Use of Animals (protocol number 4356/2022-220, date of approval–February 2022), which was approved by the State Veterinary and Food Administration of the Slovak Republic and by the Ethics Commission of the University of Veterinary Medicine and Pharmacy in Košice (Slovakia). The animals were handled in a humane manner in accordance with the guidelines established by the relevant commission. All applicable international, national, and institutional guidelines for the care and use of animals were followed. All animal owners agreed to participate in the study.

**Data Availability Statement:** All existing data are listed in the manuscript.

**Conflicts of Interest:** The authors have no competing interests to declare that are relevant to the content of this article. The authors declare they have no relevant financial or non-financial interests to disclose.

## Appendix A

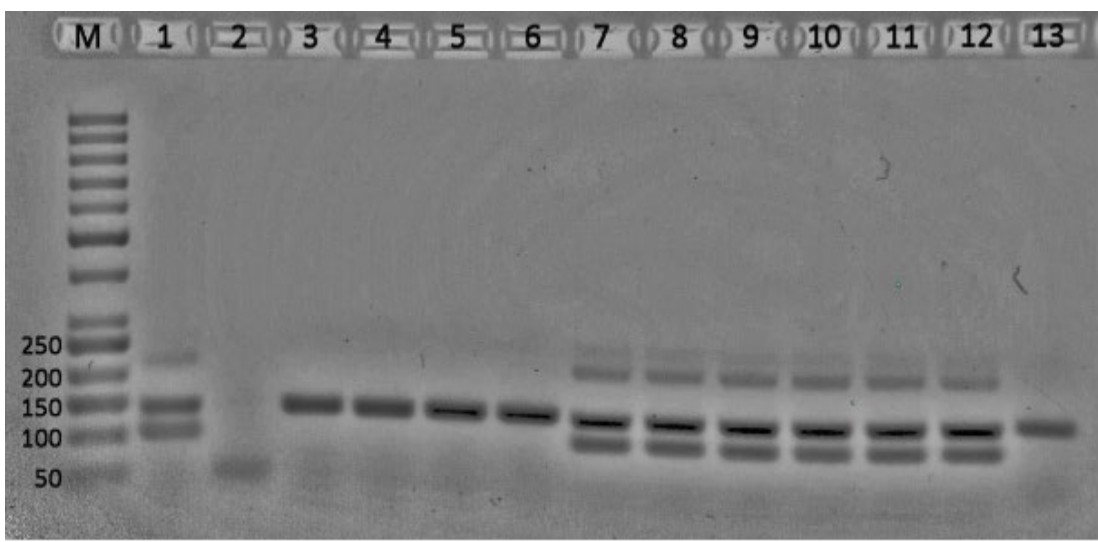

**Figure A1.** Detection of 16S rRNA, *eap* and, *nuc* genes in *S. aureus* isolates using mPCR. M—standard GeneRuler 50 bp DNA ladder; 1—control strain *S. aureus*; 2—negative control without DNA; 3 to 13—*S. aureus* isolates positive for 16S rRNA (141 bp); 7 to 12—isolates positive for gene *eap* (230 bp) and *nuc* (103 bp).

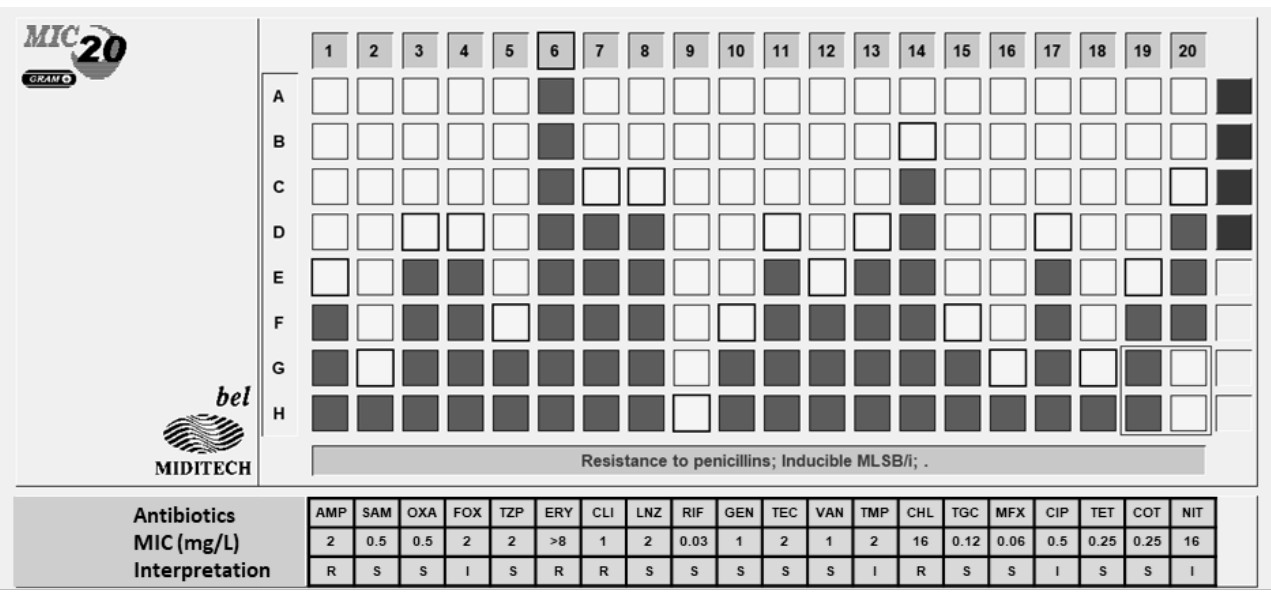

| Antibiotics | AMP | SAM | OXA | FOX | TZP | ERY | CLI | LNZ | RIF | GEN | TEC | VAN | TMP | CHL | TGC | MFX | CIP | TET | COT | NIT |
|---|---|---|---|---|---|---|---|---|---|---|---|---|---|---|---|---|---|---|---|---|
| MIC (mg/L) | 2 | 0.5 | 0.5 | 2 | 2 | >8 | 1 | 2 | 0.03 | 1 | 2 | 1 | 2 | 16 | 0.12 | 0.06 | 0.5 | 0.25 | 0.25 | 16 |
| Interpretation | R | S | S | I | S | R | R | S | S | S | S | S | I | R | S | S | I | S | S | I |

**Figure A2.** MIC results of *S. aureus* isolate (sample 10) and analysis of resistance mechanisms using the MIDITECH system. S—sensitive; I—intermediate; R—resistant; MICxG—geometric mean MIC.

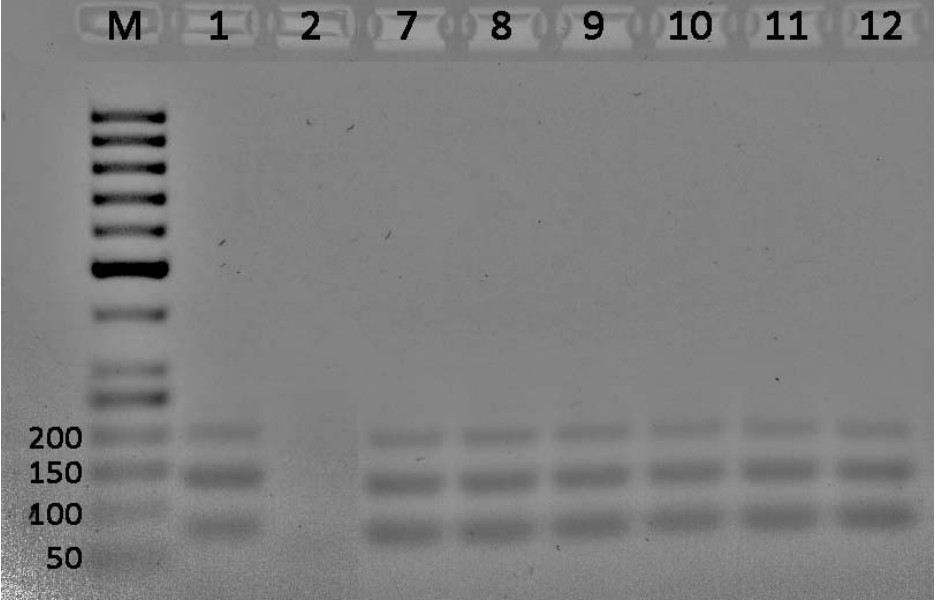

**Figure A3.** Detection of *icaA*, *icaB*, and *icaC* genes in *S. aureus* isolates by using mPCR. M—standard GeneRuler 50 bp DNA ladder; 1—control strain *S. aureus* CCM 4223; 2—negative control without DNA; 7 to 12—isolates positive for *icaA* (75 bp), *icaB* (141 bp), and *icaC* (209 bp).

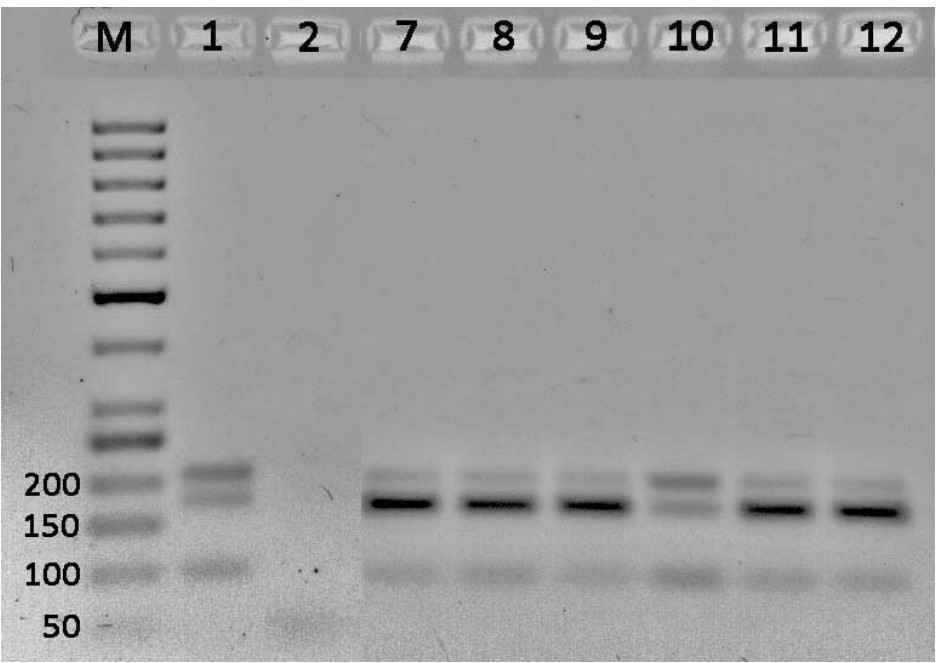

**Figure A4.** Detection of *agrA*, *srtA*, and *icaD* genes in *S. aureus* isolates by using mPCR. M—standard GeneRuler 50 bp DNA ladder; 1—control strain *S. aureus* CCM 4223; 2—negative control without DNA; 7 to 12—isolates positive for *agrA* (96 bp), *srtA* (183 bp), and *icaD* (211 bp).

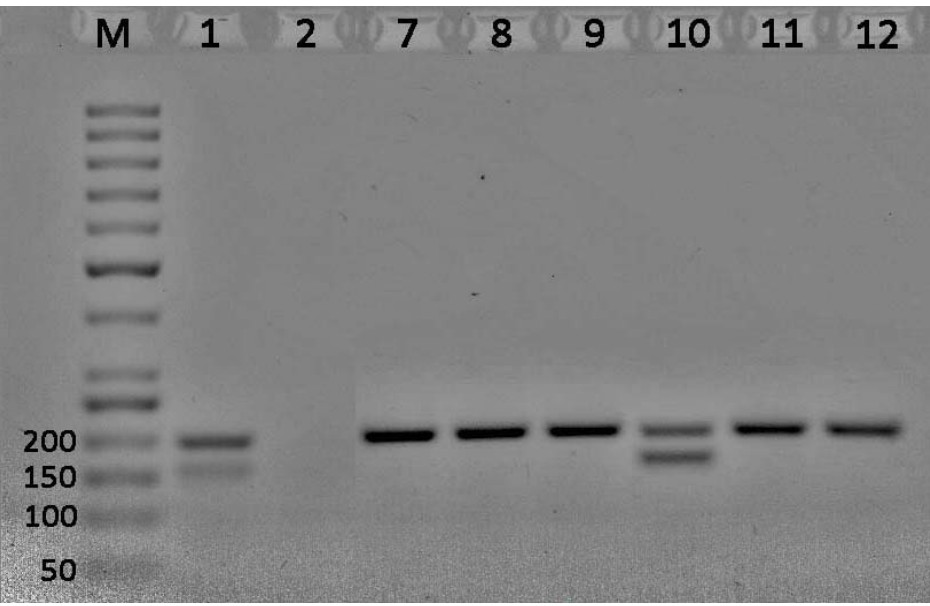

**Figure A5.** Detection of *fnbA* and *fnbB* genes in *S. aureus* isolates by using mPCR. M—standard GeneRuler 50 bp DNA ladder; 1—control strain *S. aureus* CCM 4223; 2—negative control without DNA; 7 to 12—isolates positive for *fnbA* gene (192 bp); 10—isolate positive for *fnbB* gene (156 bp).

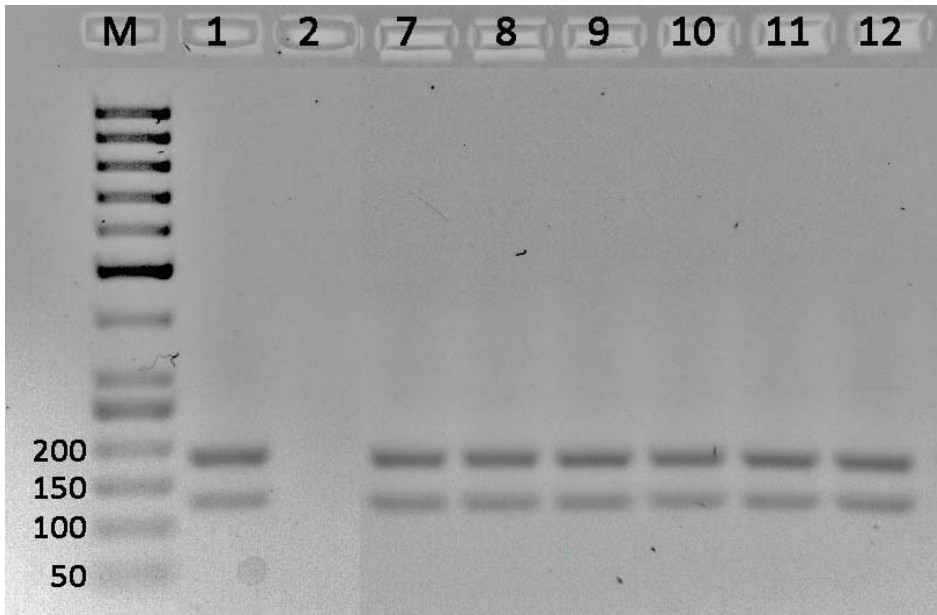

**Figure A6.** Detection of *clfA* and *clfB* genes in *S. aureus* isolates by using mPCR. M—standard GeneRuler 50 bp DNA ladder; 1—control strain *S. aureus* CCM 4223; 2—negative control without DNA; 7 to 12—isolates positive for *clfA* gene (180 bp) and *clfB* gene (120 bp).

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
