# Peer review of "Resistant S. aureus Isolates Capable of Producing Biofilm from the Milk of Dairy Cows with Subclinical Mastitis in Slovakia"

_agriculture, doi:10.3390/agriculture14040571_

Round 1

Reviewer 1 Report

Comments and Suggestions for Authors

At the abstract:

Please rephase the following "The study's objectives were to confirm S. au-13 reus, assess their antibiotic resistance, identify methicillin resistance genes, and verify biofilm for-14 mation from bovine mastitis samples using multiplex PCR (mPCR)." 
mPCR detect biofilm associated genes, it does not verify biofilm formation. Biofilm formation capability was detected by cristal violet

Introduction

The correct form for a manuscript is "4.117.031" rather than "4117031"

For a manuscript, the correct format is "67.0%" or "5.00%" rather than "67% 5%"

The following pharagraphs are not needed in the introduction since they are out of the scope of the manuscript and thus, the MLSB genes were not evaluated.

"The antimicrobial activity spectrum of MLSB antibiotics includes Gram-positive bac-92 teria (streptococci, staphylococci), Gram-negative bacteria (Bordetella pertussis, Campylo-93 bacter spp., Helicobacter spp., Legionella spp., Moraxella catarrhalis), anaerobe bacteria, intra-94 cellular pathogens (Chlamydia spp. and Rickettsia spp.) and Mycobacterium avium [6]. 95

The risk associated with MLSB-resistant S. aureus may lead to the failure of treatment 96 with erythromycin and clindamycin, which are often used to treat infections of the skin 97 and soft tissues [18].
MLSB resistance to antibiotics is associated with three main mechanisms: methyla-99 tion of rRNA, active efflux, and enzymatic inactivation. The most common genes are erm, 100 which encode rRNA methylases, resulting in the target modification of these antimicrobial 101 agents. It has been identified in more than 42 distinct erm genes. Bacteria possessing erm 102 genes exhibit cross-resistance to MLSB antimicrobial agents. On the contrary, the genes 103 encoding pumps for active efflux (msrA and lsa) or enzymes for drug inactivation (lnu and 104 mphC) confer resistance only to particular antibiotics. Based on the mechanisms of re-105 sistance, various resistant phenotypes are expressed. ΜLSB (constitutive or inducible) is 106 the most prevalent phenotype of staphylococci, associated with the presence mainly of 107 ermA and ermC genes. The MSB phenotype linked to the msrA gene follows them. In live-108 stock S. aureus strains, such as CC 398, other genes such as ermT, lnuA, lsaE and mphC 109 genes are detected [6]."

In the introduction, the following is not needed and although t
he objective is not always present at the end of the manuscript introduction. It is common to include the study aims or objectives towards the end of the introduction section

"During the final stage of biofilm formation, dispersion of some cells by hydrolysis of 144 exopolysaccharide (EPS) occurs. The released cells are converted to the planktonic form 145 and are able to create a base for new biofilm in another location. The last phase process is 146 controlled by the system Agr referred to as the quorum-sensing system (QS) which regu-147 lates expression of genes encoding RNA regulation molecules (RNAIII) in a manner de-148 pendent on the density of cells and accumulation of autoinductors acting as signal mole-149 cules (AIP – autoinductor peptide). Increased density of bacterial cells results in increased 150 concentration of AIP and reaching the threshold concentration activates the components 151 of Agr, which initiate transcription of δ-haemolysin and effector RNAIII from promotor P3. Synthesis of RNAIII results in reduced production of surface proteins and increased 153 production of exotoxins. The QS system is the basic system by which neighboring bacteria 154 exchange information and, once colonized, react genetically as a group to the host's im-155 mune response [27]."

In this regard. The introduction is somewhat bloated and could be more short and to the point.

Results

Please, remove the following
"This section may be divided by subheadings. It should provide a concise and precise 236 description of the experimental results, their interpretation, as well as the experimental 237 conclusions that can be drawn."

PCR gels picture are not needed since they do not present any resuts are just visual confimation that the PCR worked as intended. They could be added as supplementary material. Same for the MIC figure.

Discussion

Since the manuscript did not evaluated the "aminoglycosides (aac(6′)/aph(2″), aph(3′)-IIIa, ant(4′)-Ia) and MLSB (erm(A), 376 msr(A), lun(A))" genes, a discussion around it is out of the scope. Same for the MLSB since the authors have a phenotype that could or not be associated with MLSB.

Author Response

Revision according review report 1

Dear Reviewer

Thank you for your valuable comments, which can improve and repair our manuscript. Every change according your recommendation is highlighted (in green) in the manuscript. Other highlighted parts are changes according to the recommendations of other reviewers or editors (grey).

Comment of reviewer: Please rephase the following "The study's objectives were to confirm S. aureus, assess their antibiotic resistance, identify methicillin resistance genes, and verify biofilm formation from bovine mastitis samples using multiplex PCR (mPCR)." 
mPCR detect biofilm associated genes, it does not verify biofilm formation. Biofilm formation capability was detected by cristal violet
Answer: Thank you for your notes. We changed it according your recommendation.  Repaired sentence was replaced in the abstract (see line 12-14).

Comment of reviewer: Introduction

The correct form for a manuscript is "4.117.031" rather than "4117031"

For a manuscript, the correct format is "67.0%" or "5.00%" rather than "67% 5%"

Answer: Thank you for your notes. We changed it according your recommendation  (See lines 47-57)

Comment of reviewer:

The following paragraphs are not needed in the introduction since they are out of the scope of the manuscript and thus, the MLSB genes were not evaluated.

The antimicrobial activity spectrum of MLSB antibiotics includes Gram-positive bac-92 teria (streptococci, staphylococci), Gram-negative bacteria (Bordetella pertussis, Campylo-93 bacter spp., Helicobacter spp., Legionella spp., Moraxella catarrhalis), anaerobe bacteria, intra-94 cellular pathogens (Chlamydia spp. and Rickettsia spp.) and Mycobacterium avium [6]. 95

The risk associated with MLSB-resistant S. aureus may lead to the failure of treatment 96 with erythromycin and clindamycin, which are often used to treat infections of the skin 97 and soft tissues [18].
MLSB resistance to antibiotics is associated with three main mechanisms: methyla-99 tion of rRNA, active efflux, and enzymatic inactivation. The most common genes are erm, 100 which encode rRNA methylases, resulting in the target modification of these antimicrobial 101 agents. It has been identified in more than 42 distinct erm genes. Bacteria possessing erm 102 genes exhibit cross-resistance to MLSB antimicrobial agents. On the contrary, the genes 103 encoding pumps for active efflux (msrA and lsa) or enzymes for drug inactivation (lnu and 104 mphC) confer resistance only to particular antibiotics. Based on the mechanisms of re-105 sistance, various resistant phenotypes are expressed. ΜLSB (constitutive or inducible) is 106 the most prevalent phenotype of staphylococci, associated with the presence mainly of 107 ermA and ermC genes. The MSB phenotype linked to the msrA gene follows them. In live-108 stock S. aureus strains, such as CC 398, other genes such as ermT, lnuA, lsaE and mphC 109 genes are detected [6]."
Answer: Thank you for your notes. We removed this paragraph from the introduction according to your recommendations.

Comment of reviewer:

In the introduction, the following is not needed and although the objective is not always present at the end of the manuscript introduction. It is common to include the study aims or objectives towards the end of the introduction section

"During the final stage of biofilm formation, dispersion of some cells by hydrolysis of 144 exopolysaccharide (EPS) occurs. The released cells are converted to the planktonic form 145 and are able to create a base for new biofilm in another location. The last phase process is 146 controlled by the system Agr referred to as the quorum-sensing system (QS) which regu-147 lates expression of genes encoding RNA regulation molecules (RNAIII) in a manner de-148 pendent on the density of cells and accumulation of autoinductors acting as signal mole-149 cules (AIP – autoinductor peptide). Increased density of bacterial cells results in increased 150 concentration of AIP and reaching the threshold concentration activates the components 151 of Agr, which initiate transcription of δ-haemolysin and effector RNAIII from promotor P3. Synthesis of RNAIII results in reduced production of surface proteins and increased 153 production of exotoxins. The QS system is the basic system by which neighboring bacteria 154 exchange information and, once colonized, react genetically as a group to the host's im-155 mune response [27]."

In this regard. The introduction is somewhat bloated and could be more short and to the point.

Answer: Thank you for your notes. We removed this paragraph from the introduction according to your recommendations.

 Comment of reviewer:

Results

Please, remove the following
"This section may be divided by subheadings. It should provide a concise and precise 236 description of the experimental results, their interpretation, as well as the experimental 237 conclusions that can be drawn."
Answer: Thank you for your notes. We removed this paragraph from the manuscript according to your recommendations.

 Comment of reviewer:

PCR gels picture are not needed since they do not present any results are just visual confirmation that the PCR worked as intended. They could be added as supplementary material. Same for the MIC figure.

Answer: Thank you for your notes. We removed these figures from the main part of the manuscript to the Appendix, according to your recommendations.

 Comment of reviewer: Discussion

Since the manuscript did not evaluated the "aminoglycosides (aac(6′)/aph(2″), aph(3′)-IIIa, ant(4′)-Ia) and MLSB (erm(A), 376 msr(A), lun(A))" genes, a discussion around it is out of the scope. Same for the MLSB since the authors have a phenotype that could or not be associated with MLSB.

Answer: Thank you for your notes. We removed this paragraph from the manuscript according to your recommendations.

Sincerely

Reviewer 2 Report

Comments and Suggestions for Authors

The topic of manuscript entitled: “Prevalence of Resistant S. aureus Isolates Capable of Producing Biofilm from the Milk of Dairy Cows with Subclinical Mastitis in Slovakia” is interesting. Antibiotic resistance is a trend that represents one of the greatest threats to public health worldwide. Its scale is constantly growing, causing the exhaustion of therapeutic options, causing an increase in mortality from infections caused by resistant microorganisms.. The manuscript presents the topics in an orderly and logical manner. The aim of the paper is clearly formulated. The title and abstract correspond to the content of the paper. The results of the research are presented in the right organized way. The selection of sources and literature is current and complete.

Reviewer's suggestion

The title of the article indicates that the milk was collected from cows suffering from subclinical mastitis, while the Materials and Methods section lacks information on the origin of the S. aureus strains tested and the diagnosis of mastitis:

-       on what basis was subclinical mastitis diagnosed?

-       what level of somatic cells was found in the milk?

-       is the health history of the cows known? e.g., were there previous cases of mastitis in the same lactation in the cows from which milk was collected? Has treatment been applied to these animals?

Line 40: …also   isolated…  - without space

Line 160: NK-test – no explanation of the abbreviation

Line 246, 341: S. aureus not Staphylococcus aureus

Line 338: S. aureus - italics

Line 386: all not al

Author Response

Revision according review report 2

Dear Reviewer

Thank you for your valuable comments, which can improve and repair our manuscript. Every change according your recommendation is highlighted (in turquoise) in the manuscript. Other highlighted parts are changes according to the recommendations of other reviewers or editors (grey).

 Reviewer's suggestion

The title of the article indicates that the milk was collected from cows suffering from subclinical mastitis, while the Materials and Methods section lacks information on the origin of the S. aureus strains tested and the diagnosis of mastitis:

On what basis was subclinical mastitis diagnosed? What level of somatic cells was found in the milk?

Is the health history of the cows known? e.g., were there previous cases of mastitis in the same lactation in the cows from which milk was collected? Has treatment been applied to these animals?

Answer: Thank you for your notes. We add missing information to the chapter materials and methods (See the lines 127-135).

Milk samples (n = 215) were collected from dairy herds in Slovakia with reduced milk quality and milk production. Milk samples from dairy cows with subclinical mas-titis were positive for the NK-test (Natural Killer test) (Bioveta, Czech Republic). The dairy cows were without clinical signs of mastitis, without previous cases of mastitis in the same lactation, and without treatment. The NK test is a rapid determination used to find dairy cows suspected of having mastitis based on an increased somatic cell count and acid pH in milk. The somatic cell counts of 100,000–300,000 cells/ml (mild coagulation), 300,000–500,000 cells/ml (coagulation with mild gel formation), and 500,000–1,500,000 cells/ml (strong precipitation with gel formation) indicate a positive reaction.

Comment of reviewer:

Line 40: …also   isolated…  - without space

Line 160: NK-test – no explanation of the abbreviation

Line 246, 341: S. aureus not Staphylococcus aureus

Line 338: S. aureus - italics

Line 386: all not al

Answer: Thank you for your notes. We change these mistakes in the manuscript (See the lines – 39, 58, 129, 151, 221, 231, 276277, 282, 285, 329).

Reviewer 3 Report

Comments and Suggestions for Authors

The manuscript describes wide spectrum of pathogenicity in S. aures strains isolated from bovine mammary gland.

The title should be slightly modified since it is focused on bio-film formation ability while the manuscript describes much wider set of pathogenicity elements. Furthermore the word „prevalence“ seems inappropriate in the title because the study is focused on pathogenicity.

The introduction section offers plenty of useful data regarding bovine mastitis, its clinical forms, etiology and antibiotic drug consumption in veterinary medicine. Furthermore this part of the manuscript describes principles of either genetic or phenotypic resistance toward antibiotics and the main aspects of pathogenicity of staphylococci.

In the material and methods section authors describe methods applied to isolate and identify S. aureus, to extract DNA, detect selected genes and to confirm ability of tested strains to form bio-film. Methods are described with enough details to repeat the study.

Minor request in this part relies on the sample size quotation. Although the number of samples and strains is stated in the result section, authors are asked to state these numbers in the Material and method section.

Results are clearly presented as text and self-explanatory tables and figures in an unambiguous manner. Minor requests rely on a few technical details:

·         Authors are asked to replace column title Resistance to Sensitivity in Table 4 in order to be concordant with the relying subheading in the previous section (line 255: 3.2. Antimicrobial Susceptibility Profile)

·         Font size of legends bellow figures should be edited because is larger than the title of the figures.

In the discussion section authors compare own results with the results of similar studies carried out elsewhere offering reasonable explanation when results are discordant.

Conclusion section should be reduced, rephrased and drawn from results. The word prevalence should be avoided since the research does not deal with the prevalence. From such way of presentation it is almost impossible to find evidence for the statement (line 446): The results of our study show a high prevalence of biofilm-forming S. aureus isolates. Hence authors are asked to rephrase this section.

Overall the manuscript offers a plenty of useful data regarding the pathogenicity of S. aureus from Slovak dairy farms.

Author Response

Revision according review report 3

Dear Reviewer

Thank you for your valuable comments, which can improve and repair our manuscript. Every change according your recommendation is highlighted (in yellow) in the manuscript. Other highlighted parts are changes according to the recommendations of other reviewers or editors (grey).

Comment of reviewer:

The title should be slightly modified since it is focused on bio-film formation ability while the manuscript describes much wider set of pathogenicity elements. Furthermore the word „prevalence“ seems inappropriate in the title because the study is focused on pathogenicity.

Answer: Thank you for your notes. We changed the title of manuscript to: Resistant S. aureus Isolates Capable of Producing Biofilm from the Milk of Dairy Cows with Subclinical Mastitis in Slovakia

Comment of reviewer:

Minor request in this part relies on the sample size quotation. Although the number of samples and strains is stated in the result section, authors are asked to state these numbers in the Material and method section.

Answer: Thank you for your notes. We add it to the chapter materials and methods (see line 127).

Comment of reviewer:

Results are clearly presented as text and self-explanatory tables and figures in an unambiguous manner. Minor requests rely on a few technical details:

Authors are asked to replace column title Resistance to Sensitivity in Table 4 in order to be concordant with the relying subheading in the previous section (line 255: 3.2. Antimicrobial Susceptibility Profile)

Answer: Thank you, we add it to the table (see line 245).

Comment of reviewer:

Font size of legends bellow figures should be edited because is larger than the title of the figures.

Answer: Thank you, we changed it after each figure. Some of them are in Appendix according the recommendation of other reviewer (see pages 11-14).

 Comment of reviewer:

Conclusion section should be reduced, rephrased and drawn from results. The word prevalence should be avoided since the research does not deal with the prevalence. From such way of presentation it is almost impossible to find evidence for the statement (line 446): The results of our study show a high prevalence of biofilm-forming S. aureus isolates. Hence authors are asked to rephrase this section.

Overall the manuscript offers a plenty of useful data regarding the pathogenicity of S. aureus from Slovak dairy farms.

Answer: Thank you for your notes. We changed our conclusion according to your recommendations. We rephrased some sentences and shortened them according to our results.